# Distinct astrocytic modulatory roles in sensory transmission during sleep, wakefulness, and arousal states in freely moving mice

Fushun Wang [1,2,20] ✉, Wei Wang[1,3,20], Simeng Gu[1,4,20], Dan Qi [5,20], Nathan A. Smith[2,6,7], Weiguo Peng[1], Wei Dong [8], Jiajin Yuan[2], Binbin Zhao[9], Ying Mao [10], Peng Cao [11], Qing Richard Lu [12,13], Lee A. Shapiro [14] ✉, S. Stephen Yi [15,16,17] ✉, Erxi Wu [5,15,18] ✉ & Jason H. Huang [5,18,19] ✉

Despite extensive research on astrocytic $Ca^{2+}$ in synaptic transmission, its contribution to the modulation of sensory transmission during different brain states remains largely unknown. Here, by using two-photon microscopy and whole-cell recordings, we show two distinct astrocytic $Ca^{2+}$ signals in the murine barrel cortex: a small, long-lasting $Ca^{2+}$ increase during sleep and a large, widespread but short-lasting $Ca^{2+}$ spike when aroused. The large $Ca^{2+}$ wave in aroused mice was inositol trisphosphate (IP3)-dependent, evoked by the locus coeruleus-norepinephrine system, and enhanced sensory input, contributing to reliable sensory transmission. However, the small $Ca^{2+}$ transient was IP3-independent and contributed to decreased extracellular $K^+$, hyperpolarization of the neurons, and suppression of sensory transmission. These events respond to different pharmacological inputs and contribute to distinct sleep and arousal functions by modulating the efficacy of sensory transmission. Together, our data demonstrate an important function for astrocytes in sleep and arousal states via astrocytic $Ca^{2+}$ waves.

It is increasingly appreciated that rather than being mere supporting cells as purely nonfunctional glues for neurons, glial cells may play active roles in sleep, arousal, and cognitive processing[1–5]. More than a century ago, Cajal indicated that astrocytes might play a critical role in sleep by extending their processes into the synapse to block sensory transmission[6]. Later electron microscopy studies demonstrated that sleep deprivation led to extensions of astrocytic processes towards the synaptic cleft[7], supporting Cajal's idea. In recent decades, many more studies have examined the functional role of astrocytes in cognition during sleep and waking states[8–18]. In addition, astrocytes have also been suggested to affect circadian rhythm[11] and control slow oscillation sleep[15,19], as well as neural circuits within distinct behavioral states[17,20,21], which suggests a direct functional role for astrocytes in sleep and arousal. However, the direct functional astrocytic role in emotional arousal states is not clear.

Sleep, wakefulness, and arousal represent distinct alert states characterized by unique emotional activity profiles and different cognitive functions[22]. During periods of high arousal states, brain electroencephalogram (EEG) recordings show desynchronized, high-frequency, low-amplitude patterns, which are modulated by short bursts of high-frequency neuromodulatory activity[23]. The sleep-wake cycle and emotional arousal states are primarily regulated by neuromodulators[12]. Astrocytes have a unique capability to amplify and extend the effects of neuromodulators over a large population of synapses, making them a particularly intriguing component of the brain's signaling system[12]. Previous findings, including our own, also showed that sensory input can induce local astrocytic $Ca^{2+}$ waves[17,24] and can trigger widespread $Ca^{2+}$ signals in the cerebral cortex mediated by α1 adrenergic receptor activation in waking states[25]. The locus coeruleus-norepinephrine (LC-NE) projection is recognized to play a

A full list of affiliations appears at the end of the paper. ✉ e-mail: 13814541138@163.com; lshapiro@tamu.edu; Stephen.yi@austin.utexas.edu; Erxi.Wu@bswhealth.org; Jason.Huang@bswhealth.org

major role in regulating the behavioral state and state-dependent processing of sensory information[26,27]. However, the functional role of astrocytes in sleep, arousal, and their relationship with monoamine neuromodulators are not clear.

In this study, we incorporate genetically encoded Ca[2+] indicators (GCaMP6f) into this circuitry, enabling the identification of unique astrocytic Ca[2+] signals during sleep, arousal, and sensory transmission[25]. We conduct functional studies in live mice, incorporating dual whole-cell recordings in single neurons and local field potential (LFP) recordings in layer II/III of the barrel cortex. We show that astrocytic signaling is differentially regulated by the previously described function of norepinephrine in modulating sleep and arousal, as well as sensory input[28], demonstrating a

fundamental role for astrocytes in sensory processing and emotional arousal.

## Results

### Astrocytic processes in the barrel cortex exhibit two distinct Ca[2+] signals at sensory input

To probe the function of astrocytic Ca[2+] signaling during different behavioral states, a genetically encoded calcium indicator (GCaMP6f) using a GfaABC1D promoter was delivered selectively into astrocytes using adeno-associated virus (AAV) vectors without detectable astrogliosis[29], enabling detection of Ca[2+] transients[2] (Fig. 1A, B). Two weeks after virus microinjections, GCaMP6f-expressing astrocytes were monitored under two-photon microscopy. On the day of the

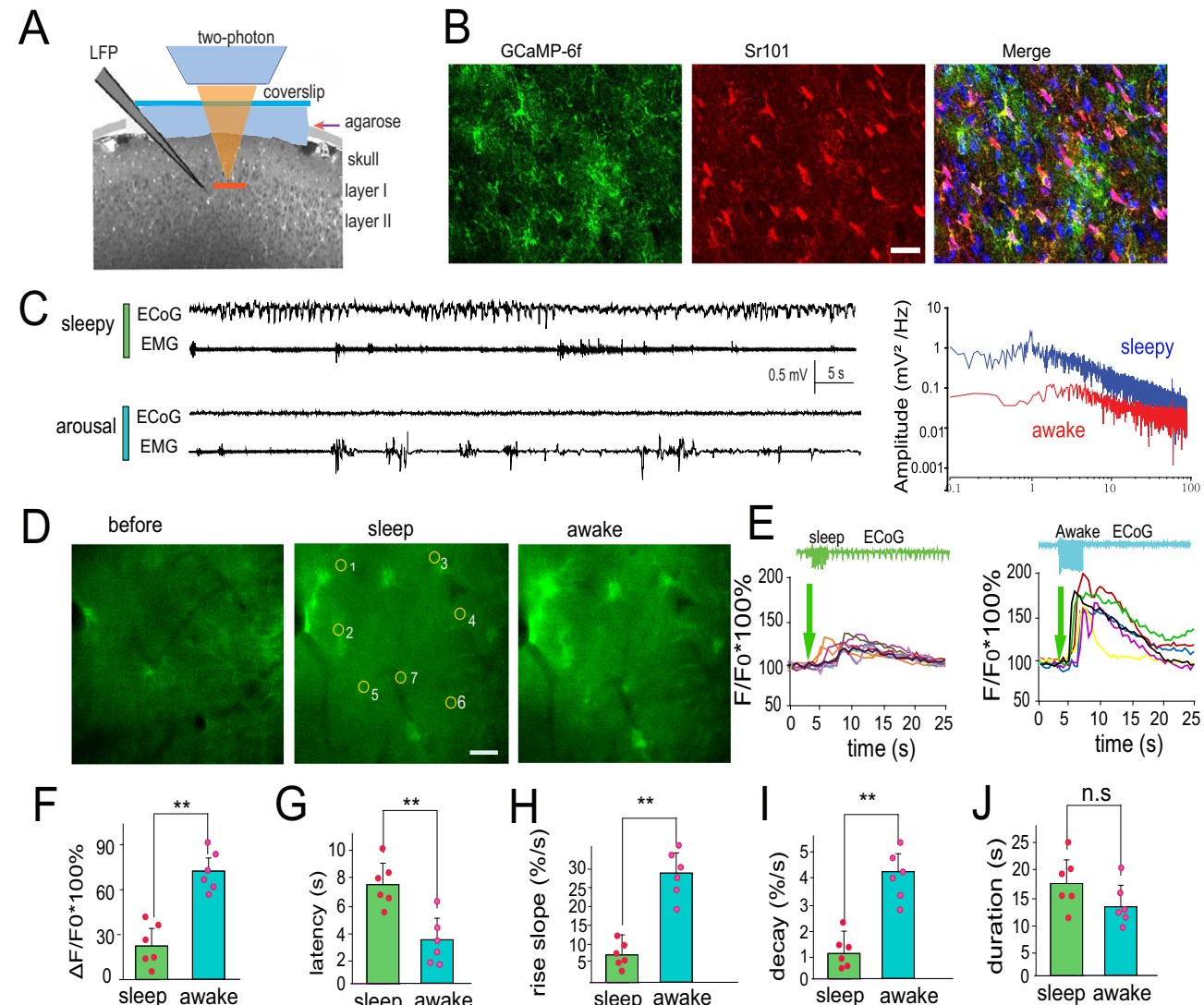

**Fig. 1 | Two distinct Ca[2+] waves at astrocytic fine processes are characterized for the sleep and awake states. A** A schematic drawing depicting the experimental setup combining two-photon imaging of astrocytic Ca[2+] signaling in the barrel cortex with in vivo whole-cell recordings in single neurons and local field recordings in awake behaving GCaMP6f transgenic mice. **B** Representative fluorescence photomicrographs of the barrel cortex displaying the expression of GCaMP6f (left panel), SR101-loaded astrocytes (middle panel), and merged images (right panel) illustrating that some of the GCaMP6f-expressing astrocytes were colabeled with SR101-loaded astrocytes. Scale bar = 10 μm. **C** Representative recordings of EEG and EMG and power spectrum analyses during sleep and in awake mice (blue as sleep; red as arousal). Upper traces are representative ECoG traces from whisker stimulation. Awake states were shown as no apparent evoked ECoG response and strong

EMG activity. **D** Representative fluorescence changes showing that whisker stimulation induced small Ca[2+] increases during sleep, scale bar = 10 μm. **E** The analyses of fluorescence changes corresponding to GCaMP6f F/F$_0$ traces at the fine processes, which are marked as circles in (**D**), showing the higher amplitude response to the stimulation in awake mice. Note the significant increase in the peak in the awake state. **F–J** Statistical data show the comparison of whisker stimulation-induced changes in Ca[2+] transients by measuring ΔF/F$_0$ (**F**), latency (**G**), rise slope (**H**), decay (**I**), and duration (**J**), confirming the differential firing dynamics during sleep and awake states (the data are shown as the mean ± SD, **paired t-test, $P < 0.001$; n.s not significant, $P = 0.161$. The dots represent the mean value in each mouse, $n = 6$ mice in each group).

experiment, a cranial window was prepared for $Ca^{2+}$ imaging with two-photon microscopy, as well as in vivo whole-cell recordings of single neurons and local field potentials (LFPs) for groups of neurons in layer II/III of the barrel cortex (Fig. 1A)[11]. It was reported that sleep-wake brain states and motor behaviors can be identified as four states with different levels of brain arousal and motor activity[30]. Sleep was assessed both qualitatively as periods in which the mice closed their eyes and quantitatively using spectral analysis of electroencephalography (EEG) recordings that demonstrated slow waves of 0.5–4 Hz for EEG recording[31] (Fig. 1C), concurrent with relatively low amplitude electromyography (EMG) recordings in the neck (Fig. S1). The waking state was characterized by eyes being open and spontaneous whisker movement, and these behaviors were accompanied by typical EMG recordings (Fig. 1C). In both sleep and waking states, low-frequency whisker stimulation (5 Hz) significantly increased astrocytic $Ca^{2+}$ transients. However, whisker stimulation-induced $Ca^{2+}$ transients during sleep were significantly smaller than those during the awake state, and peak fluorescence increased $26.2 \pm 12.6\%$ for whisker stimulation during sleep compared with $74.6 \pm 13.2\%$ for this same stimulation during awake states ($P < 0.01$, paired t-test, $n = 54$ regions of interest (ROIs) in 6 mice, Fig. 1D–F; Supplementary Movie 1). In awake mice, the larger $Ca^{2+}$ transients had a faster latency, $3.5 \pm 2.3$ s compared with $7.8 \pm 1.8$ s in the sleep state ($n = 54$ ROIs in 6 mice, Fig. 1G), and a faster increase in fluorescence ($26.7 \pm 4.3\%$ per second compared with $5.1 \pm 2.1\%$ per second in the sleep state, $P < 0.01$, Fig. 1H). In addition, the decay was also faster for the astrocytic $Ca^{2+}$ signals in awake mice ($4.1 \pm 1.1\%/s$ compared with $1.2 \pm 0.8\%/s$ during sleep, $P < 0.01$, paired t-test, Fig. 1I), whereas there were no significant differences in the decay durations during sleep vs in awake mice (with $14.1 \pm 3.7$ s compared with $18.3 \pm 4.6$ s for sleep, paired t-test, $P = 0.161$, Fig. 1J). Together, these results suggest that astrocytic processes exhibit two distinct $Ca^{2+}$ signals that respond to environmental stimulation in the barrel cortex.

## Noradrenergic activity from the LC drives the switch between the two distinct $Ca^{2+}$ signals

Previous studies have shown that astrocytic $Ca^{2+}$ transients require norepinephrine (NE) from the LC to enable system priming[31,32]. To confirm these findings, we stimulated the LC and compared the two kinds of $Ca^{2+}$ transients and found that LC stimulation changed the small $Ca^{2+}$ transients during sleep to larger $Ca^{2+}$ transients and caused powerful arousal in the mice (fluorescence transient was $27.8 \pm 3.9\%$ for whisker stimulation before LC stimulation; after LC stimulation, $Ca^{2+}$ transients increased by $64.01 \pm 6.47\%$, $n = 44$ ROIs in 5 mice, **paired t-test, $P < 0.01$) (Fig. 2A, B). To confirm that LC stimulation was mediated through NE signaling, NE (100 μM) was applied to the barrel cortex during sleep, followed by whisker stimulation. The whisker stimulation in the presence of NE resulted in a significant enhancement of $Ca^{2+}$ signaling from $27.3 \pm 4.9\%$ to $65.8 \pm 6.2\%$ ($n = 59$ ROIs in 6 mice, **paired t-test, $P < 0.01$), similar to the data reported above for LC stimulation with whisker stimulation.

Because these findings appear to show the involvement of NE signaling in this effect, subsequent experiments were designed to identify specific NE receptors involved in the LC/NE-mediated induction of barrel cortex signaling using receptor antagonists. First, we introduced the alpha-1 adrenergic receptor ($\alpha_1$-AR) antagonist terazosin (100 μM), and this drug significantly blocked LC stimulation-induced $Ca^{2+}$ signaling (peak of $\Delta F/F_0$ in the presence of the drug was $34.5 \pm 6.8\%$ compared to the absence of the drug, $P < 0.01$, one-way ANOVA, $n = 61$ ROIs in 5 mice, Fig. 2D). Conversely, administration of the alpha-2 adrenergic receptor ($\alpha_2$-AR) antagonist metoprolol (10 μM) was not effective ($\Delta F/F_0$ peaked at $55.2 \pm 5.9\%$, $P < 0.01$, one-way ANOVA, $n = 58$ ROIs in 5 mice) at significantly blocking LC-induced $Ca^{2+}$ signaling. Thus, $\alpha_1$-AR but not $\alpha_2$-AR appears to be involved in the LC/NE-mediated induction of barrel cortex signaling.

## Large astrocytic $Ca^{2+}$ transients in awake mice enhance sensory transmission

As reported before, four states with different levels of brain arousal and motor activity can be identified[31]. Indeed, the behavioral states showed that there might be four kinds of states: deep sleep-sleep-quiet wakefulness-arousal states. The arousal state is characterized as eye open, whisker movement, and running the pedals. Therefore, whisker stimulation during sleep induced smaller $Ca^{2+}$ signaling that might induce even deeper sleep, whereas whisker stimulation in the waking state induced a larger $Ca^{2+}$/transient, which might lead the mouse to aroused states from quiet wakefulness (Fig. 2E). Whisker stimulation in the waking state induced behavioral arousal; consistently, whole-cell recordings have shown that LC stimulation enhanced EPSPs in vitro[33,34]. Comparisons were also performed between whisker stimulation-evoked excitatory postsynaptic potentials (EPSPs) before and after LC stimulation and revealed that LC stimulation significantly increased the amplitude of whisker stimulation-induced EPSPs (from $0.768 \pm 0.33$ mV for LFP recordings for whisker stimulation before LC stimulation, compared with $1.38 \pm 0.46$ mV for whisker stimulation after LC stimulation, $n = 5$ mice, one-way ANOVA, $P < 0.01$, Fig. 2F). Interestingly, when the mice returned to the sleep state, the EPSPs were again smaller (Fig. S2).

To further test the functional effects of the larger astrocytic $Ca^{2+}$ on sensory transmission in the waking state, we used traditional agonists to induce $Ca^{2+}$ as shown in our previous reports[22,35]. Consistently, the agonists induced large $Ca^{2+}$ transients in awake mice and induced aroused behavioral states with more locomotion (Fig. 3A, B). Then, we tested the effects of larger astrocytic $Ca^{2+}$ in EPSPs in waking states. In vivo whole-cell recordings showed that EPSPs were increased after ATP (100 μM)-induced astrocytic $Ca^{2+}$ transients, from $16.2 \pm 1.4$ mV to $21.5 \pm 1.5$ mV ($132.7 \pm 10.8\%$ increase; **, $P < 0.01$, $n = 7$ mice, Fig. 3C, D). The EPSPs recorded with LFP also increased from $0.95 \pm 0.107$ mV to $1.33 \pm 0.15$ mV ($140.71 \pm 13.74\%$; **, $P < 0.01$, t-test, $n = 8$ mice, Fig. 3C, D). Of note, whole-cell EPSPs occurred at $29.6 \pm 1.76$ ms after air puffing ($n = 8$ cells, one cell in each mouse), while LFP EPSPs occurred with a latency of $20.2 \pm 1.3$ ms after whisker stimulation (Fig. 3D). However, the LFP EPSPs terminated much earlier than the whole-cell recorded EPSPs ($60.41 \pm 4.6$ ms, compared with $192.28 \pm 8.16$ ms for whole-cell recordings, $n = 8$ mice, t-test, $P < 0.01$; Fig. 3D).

In addition, UTP (100 μM) induced $Ca^{2+}$ transients in astrocytes and increased the amplitude of EPSPs, which increased from $13.7 \pm 2.1$ mV before UTP treatment to $18.3 \pm 2.2$ mV after UTP treatment (*$P < 0.05$, t-test, $n = 7$ mice, Fig. 3E). To test the specificity of the induced $Ca^{2+}$ in astrocytes, we also used GFAP-Gq-DREADD-transfected mice, followed by administration of clozapine-N-oxide (CNO, 100 μM) to specifically stimulate the transfected astrocytes. Administration of CNO significantly induced $Ca^{2+}$ in astrocytes, which was $68.5 \pm 3.4\%$ with a duration of $21.3 \pm 2.6$ s ($P < 0.05$, $n = 64$ ROIs in 7 mice), and whisker stimulation induced EPSPs, which increased from $14.1 \pm 1.5$ mV before UTP to $18.5 \pm 2.2$ mV ($131.2 \pm 13.3\%$ increase; $P < 0.05$, $n = 7$ mice, Fig. 3E, F). At the same time, the membrane potential depolarized from a resting $-70.6 \pm 2.3$ mV to $-64.6 \pm 2.1$ mV ($0.91 \pm 0.16$; $P < 0.05$, $n = 7$ mice, Fig. 3G), possibly due to the extracellular $K^+$ increases[22,36]. To further determine the astroglial origin of this EPSP enhancement, agonist-induced $Ca^{2+}$ signals were studied in MrgA1+ mice[36]. In these mice, the Gq-linked MrgA1 receptor is expressed in astrocytes under the control of the glial fibrillary acidic protein (GFAP) promoter; the MrgA1 receptor can be selectively stimulated with Phe-Met-Arg-Phe-NH2 (FMRF) amide. The results showed that administration of FMRF (100 μM) significantly induced $Ca^{2+}$ in astrocytes (one-way ANOVA, $F(7, 470) = 4.7$, **$P < 0.01$, $n = 27$ ROIs in 7 mice). Most importantly, EPSP-induced neuronal firing (action potentials, APs) also increased by $55.6 \pm 26.3\%$ (one-way ANOVA, $F(3,27) = 27.34$, *$P < 0.01$, $n = 27$ ROIs in 7 mice, Fig. 3H). Therefore, the large $Ca^{2+}$ waves appear to enhance sensory responsiveness, suggesting that astrocytes act as a gain to

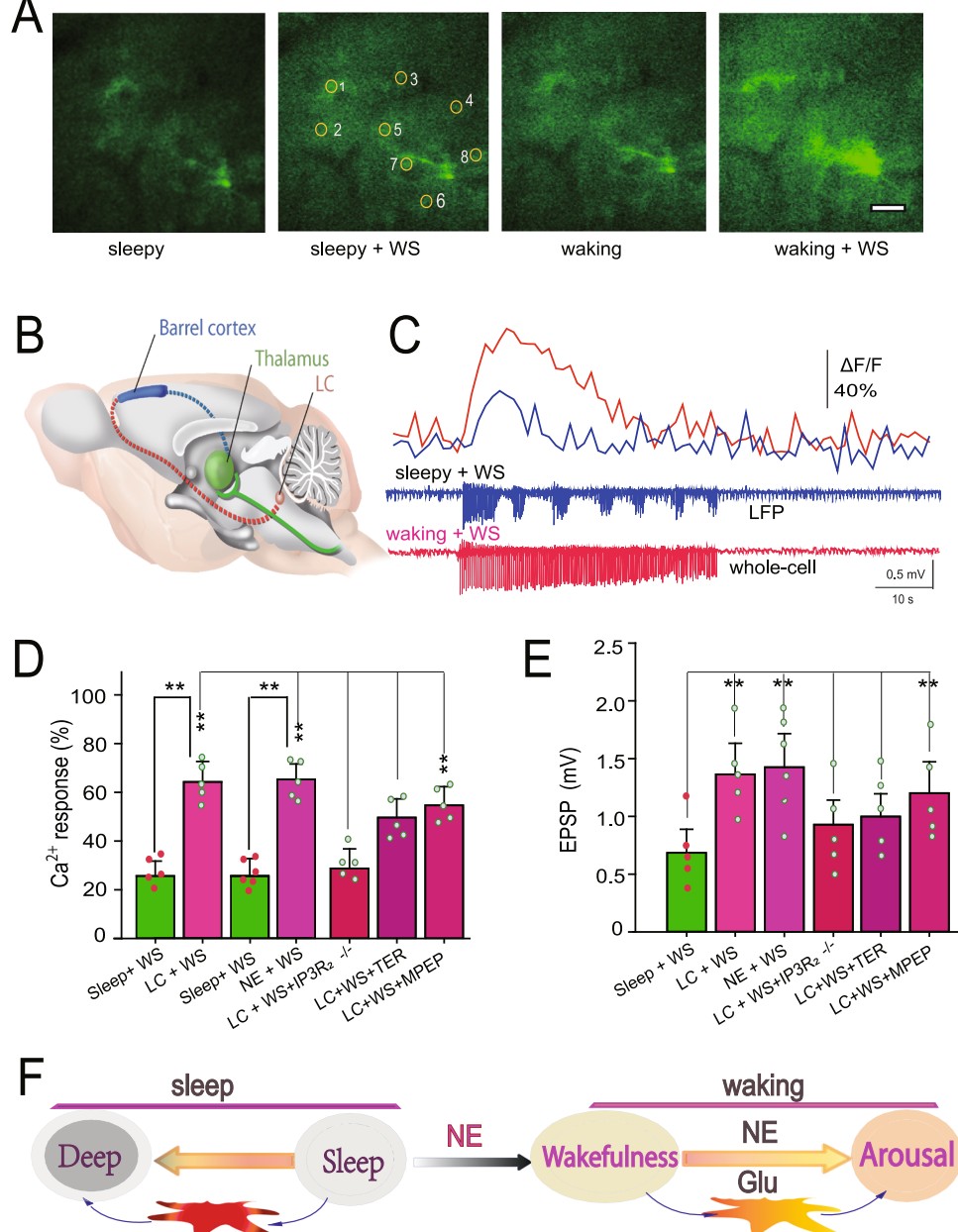

**Fig. 2 | LC stimulation of adrenergic activity switched small Ca²⁺ signaling to large waves. A** Typical fluorescent images show that LC stimulation can induce large Ca²⁺ increases after LC stimulation. Scale bar = 20 µm. **B** A schematic illustrates projections from LC stimulation (red dashed line) and whisker stimulation (green solid line and blue dashed line) to the barrel cortex. LC stimulation was generated using bipolar concentric electrodes, while whisker stimulation was generated through air puffing. **C** Typical traces show the changes in Ca²⁺ transients and LFP EPSPs. Upper traces (blue trace is from sleeping state, while the red trace is after LC stimulation) are relative changes of Ca²⁺ ($\Delta F/F_O$) in an astrocytic process located close to the recording electrode. Lower traces are LFPs recorded by the recording electrode. **D** Analysis shows comparisons of the relative changes in Ca²⁺ ($\Delta F/F_O$) in astrocytic processes upon whisker stimulation (**$P < 0.001$, paired t-test for the first two pair of groups, $n = 42$ for before and after LC stimulation; $n = 58$ for before and after NE application. For other groups, one-way ANOVA was used, F (4,296) = 227.373, $P < 0.001$, spots show the average in each mouse, $n = 5$–$6$ mice, data are shown as the mean ± SD). **E** Analysis shows comparisons of the relative changes in LFP EPSPs upon whisker stimulation (**$P < 0.01$, one-way ANOVA, $n = 5$–$6$ mice, data are shown as the mean ± SD). The x-axis labels indicate the type of stimulation used. Additional abbreviations used are as follows: TER, α1-antagonist terazosin; MPEP, mGluR5 antagonist. **F** A cartoon shows the possible mechanism of two different effects of Ca²⁺ during sleep and waking, and NE is a switch between these two states.

mediate the salience or impact of sensory flow into the neocortex by modulating their Ca²⁺ signals to induce a brain arousal state.

## Large astrocytic Ca²⁺ transients in the awake state are dependent on mGluR

The characteristics of astroglial Ca²⁺ signals induced by whisker stimulation were examined in the presence of two antagonists: mGlu5 receptor antagonist MPEP (50 µM) and mGlu2,3 receptor antagonist

LY341495 (10 µM). The resulting Ca²⁺ signal, $\Delta F/F_O$, decreased to 19.2 ± 6.6% during waking states (56 ROIs in 5 mice), as shown in Fig. 3B. However, when MPEP (50 µM) alone was used, the decrease in $\Delta F/F_O$ was not significant (Fig. 2D). This finding is consistent with a previous report indicating that the expression of mGluR5 in astrocytes is developmentally regulated and undetectable after postnatal week 3, whereas mGluR3 is expressed in astrocytes at all developmental stages[37]. This also suggests that astroglial Ca²⁺ signals in the awake

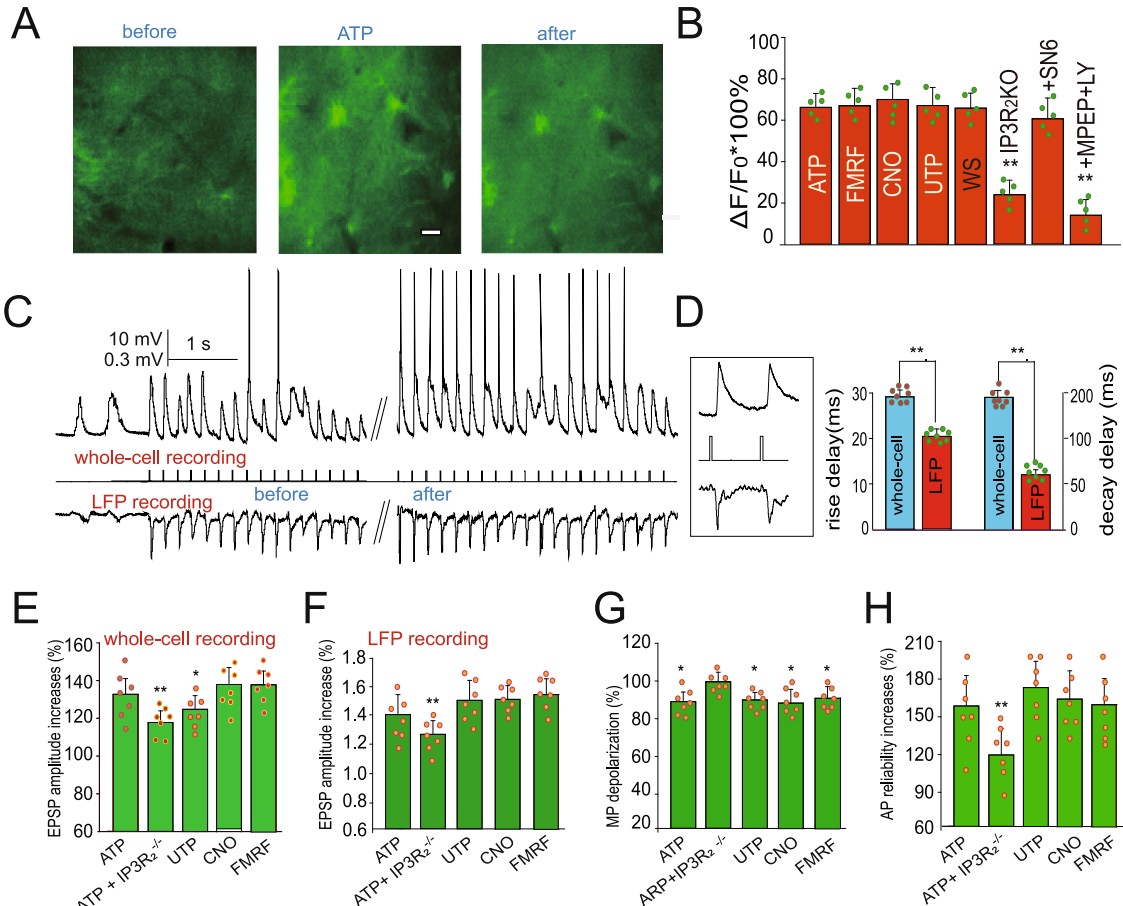

**Fig. 3 | Whisker stimulation induces large Ca$^{2+}$ transients in awake mice that are glutamate-dependent. A** Fluorescent images showing that agonist (ATP 100 μM) induced large Ca$^{2+}$ increases during waking states. Scale bar = 10 μm.
**B** Comparisons of the relative changes in Ca$^{2+}$ (ΔF/F$_0$) in astrocytes upon whisker stimulation (**$P < 0.01$, one-way ANOVA, $F_{(5, 391)} = 394.3$; data are shown as the mean ± SD; spots show the average in each mouse, $n = 5$ mice in each group). LY: mGluR2-3 antagonist LY341495. **C** Typical recordings show whisker stimulation (WS)-induced EPSP increases after agonist-induced astrocytic Ca$^{2+}$ signaling at the barrel cortex in live mice. **D** Typical traces (in the inlet) and statistical analyses show the differences in whole-cell recorded EPSPs and LFP EPSPs (**, paired t-test, $n = 8$ mice; dots show the values in each mouse; data are shown as the mean ± SD in the

bar graphs). **E, F** Statistical analysis of EPSP increases after astrocytic Ca$^{2+}$ signaling induced by agonists (data are shown as the mean ± SD; **$P < 0.01$, one-way ANOVA was used, $F_{(4,34)} = 126.7$. Spots show the averaged value of EPSPs in each mouse, $n = 7$ mice). Comparisons of the amplitude of EPSPs recorded with both whole-cell recordings (**E**) and LFP (local field potential) recordings (**F**) during sleep and waking states. **G** Analysis of membrane potential (MP) changes from in vivo whole-cell recordings showing membrane potential depolarization. Data are shown as the mean ± SD. *$P < 0.05$. **H** Analysis of the reliability of whisker stimulation-induced action potentials before and after Ca$^{2+}$ signaling. Data are shown as the mean ± SD. **$P < 0.01$.

state are mediated by glutamate-activated, metabotropic glutamate receptors (mGluRs). In mice with genetic deletion of the IP$_3$ receptor (Type 2, IP$_3$R$_2$-/-), whisker stimulation-induced Ca$^{2+}$ signals were significantly decreased to 21.6 ± 5.8% ($n = 64$ ROIs in 5 mice, Fig. 3B), demonstrating the importance of this system in regulating sensory information flow. Administration of the Na$^+$/Ca$^{2+}$ exchanger (NCX) blockers SN-6 (SN-6:2-[4-(4-nitrobenzyloxy)benzyl]thiazolidine-4-carboxylic acid ethyl ester, 50 μM) and SEA-0400 (SEA0400: 2-[4-[(2,5-difluorophenyl)methoxy]-phenoxy]-5-ethoxyaniline, 50 μM) had a modest effect on whisker stimulation-induced Ca$^{2+}$ transients, with an average peak fluorescence of 58.6 ± 7.7% ($n = 59$ ROIs in 5 mice, Fig. 3B). Thus, the large Ca$^{2+}$ transient that enhances sensory transmission in the barrel cortex appears to be dependent upon IP-dependent Ca$^{2+}$ release from the endoplasmic reticulum (ER).

**Small Ca$^{2+}$ transients block sensory transmission during sleep**
Considering that a functional role for the large Ca$^{2+}$ transients during arousal was elucidated, we next assessed the functional effects of the small Ca$^{2+}$ transients on neuronal activity and attempted to determine whether the smaller Ca$^{2+}$ transients can induce deeper sleep (Fig. 2E,

Supplementary Movie 2). We performed in vivo whole-cell recordings for neurons in layers II/III of the barrel cortex, and the results showed that the recorded neurons stereotypically oscillated between two intrinsic stable membrane potentials during sleep (1.23 ± 0.25 Hz, −73.3 ± 2.8 mV, and −60.8 ± 3.5 mV, $n = 18$ cells in 18 mice, one cell in each mouse, Fig. 4A, B, Fig. S3). Contrary to previous reports that the thalamus could block all sensory transmission to the cortex during sleep, whisker stimulation elicited EPSPs, which were assessed using both whole cell recording (mean ± SD is 14.8 ± 2.7 mV for the first ten EPSPs, $n = 8$ cells, Fig. 4C, D) and LFP recordings (averaged at 0.82 ± 0.14 mV for the first ten EPSPs, $n = 8$ mice, Fig. 4E).

We screened the effects of smaller astrocytic Ca$^{2+}$ transients on whisker stimulation-induced EPSPs by first applying the agonist ATP (100 μM). The results showed that ATP-induced Ca$^{2+}$ transients decreased whisker stimulation-induced EPSPs (13.8 ± 2.3 mV compared with 6.8 ± 0.9 mV after Ca$^{2+}$ transients, paired t-test, $P < 0.01$, $n = 8$ mice). Similarly, the EPSPs recorded in LFPs also decreased from 0.82 ± 0.18 mV to 0.38 ± 0.14 mV (paired t-test, $P < 0.01$, $n = 8$ mice, Fig. 4D). To determine the astroglial origin of this EPSP inhibition, agonist-induced Ca$^{2+}$ signals were studied in MrgA1$^+$ mice[36]. Injection

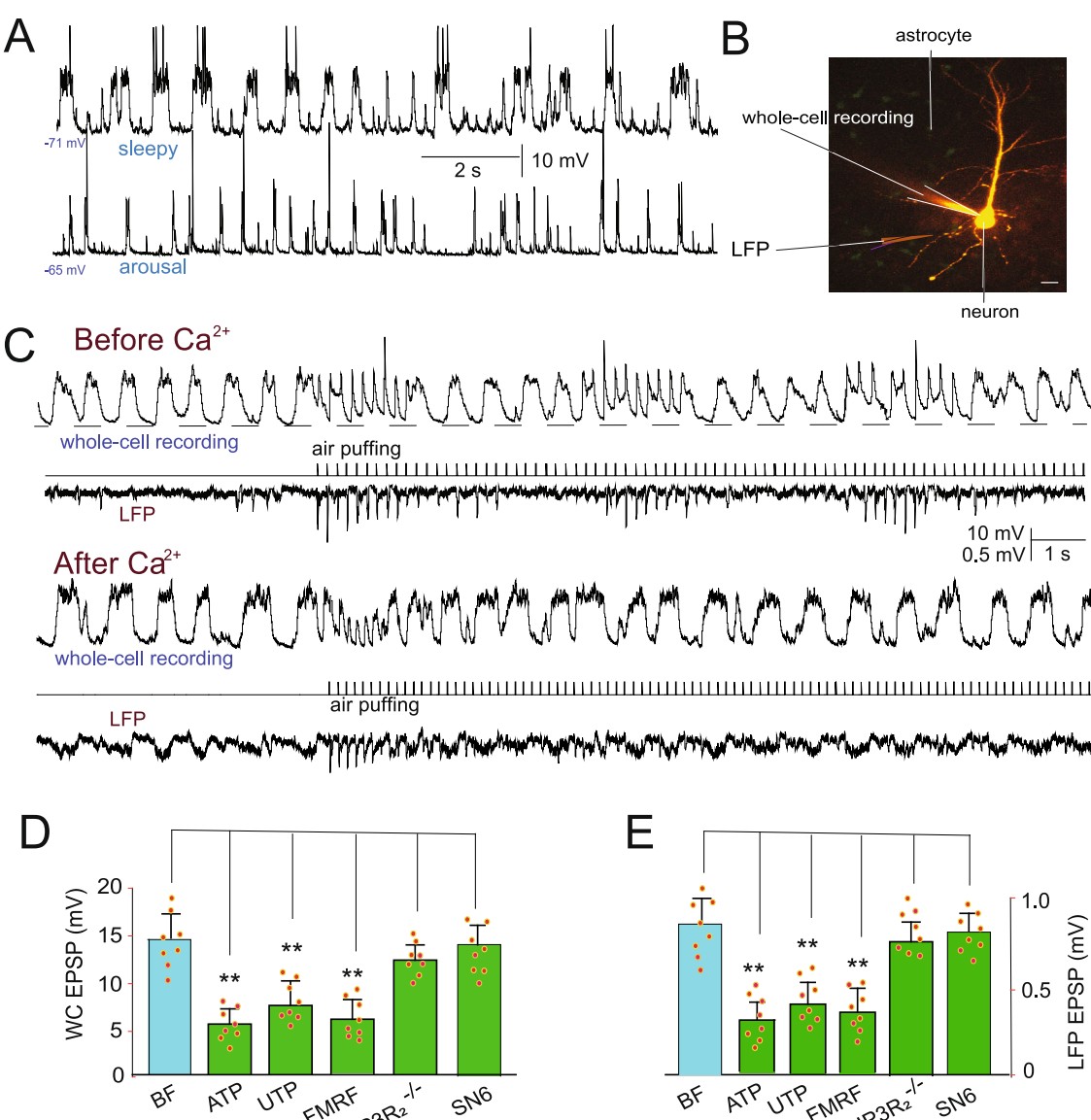

**Fig. 4 | Small Ca$^{2+}$ transients blocked sensory transmission. A** Typical recordings showing the differences in membrane potential (MP) during sleep and awake states. **B** Typical images showing a whole-cell recorded neuron (Alex 488 was dilated in intracellular solution) and images of astrocytes transfected with GCaMPf6. Scale bar = 20 μm. **C** Typical traces of whisker stimulation-induced EPSPs recorded with whole-cell recordings (Vm EPSPs) and local field recordings (LFP EPSPs) during the sleep state. **D, E** Comparisons of the amplitude of EPSPs recorded with whole-cell recordings (**D**) and LFP recordings (**E**) during sleep (**, one-way ANOVA, F (5, 47) = 27.68. Spots show the EPSP value in each mouse). The amplitude of EPSPs decreased after agonist-induced Ca$^{2+}$ signaling compared with before (**$P < 0.01$, $n = 8$ mice; BF, before; AF, after). Data are shown as the mean ± SD.

of FMRF (10 μM) evoked [Ca$^{2+}$]$_i$ transients with an average increase of 19.7 ± 4.1% during the sleep period. Furthermore, FMRF-induced [Ca$^{2+}$]$_i$ transients decreased whisker stimulation-induced EPSPs, with an amplitude decrease from 14.4 ± 2.5 mV to 7.1 ± 1.2 mV with whole-cell recording (one-way ANOVA, **$P < 0.01$, $n = 8$ mice, Fig. 4E). Therefore, these Ca$^{2+}$ signals are likely to be attributed to astrocytic function, further supporting a major role for astrocytes in mediating sleep, arousal, and sensory transmission.

**Small Ca$^{2+}$ transients during sleep depend on multiple sources**

Considering the observation of sleep and awake Ca$^{2+}$ transients, we next tested the mechanisms for the Ca$^{2+}$ transients during the sleep state. We applied glutamate receptor antagonists with ionophoresis. The results showed that the α-amino-3-hydroxy-5-methyl-4-iso-xazolepropionic acid receptor (AMPA) receptor antagonist 6-cyano-7-nitroquinoxaoine-2,3-dione (CNQX, 10 μM) had no significant effect on

the small Ca$^{2+}$ transients ($P = 0.34$, one-way ANOVA, $n = 35$ ROIs in 5 mice). The N-methyl-D-aspartate receptor (NMDA) antagonist DL-2-amino-5-phosphono-pentanoic acid (APV, 50 μM) also did not significantly alter the small transients ($P = 0.45$, one-way ANOVA, $n = 35$ ROIs in 5 mice). However, the mGlu5 receptor antagonist MPEP (50 μM) and mGlu2,3 receptor antagonist LY341495 (10 μM) significantly decreased Ca$^{2+}$ transients (mean ± SD: 10.5 ± 4.2, $P < 0.01$, one-way ANOVA, $n = 36$ ROIs in 5 mice), suggesting a role for mGLuR receptor-induced Ca$^{2+}$ release from the ER. We then used the traditional agonists ATP (adenosine 5'-triphosphate, 100 μM) and UTP (uridine 5'-tripho-sphate, 100 μM) to activate Gq receptors to determine their effects on sleeping Ca$^{2+}$ transients in astrocytes. To exclude the involvement of neurons, tetrodotoxin TTX (1 μM) was applied together with these drugs. The results suggested that these agonists could induce small astrocytic Ca$^{2+}$ transients (19.8 ± 4.5% for ATP, 17.8 ± 5.8% for UTP, $n = 34-41$ processes in 5 mice, Fig. 5B). There are typically no

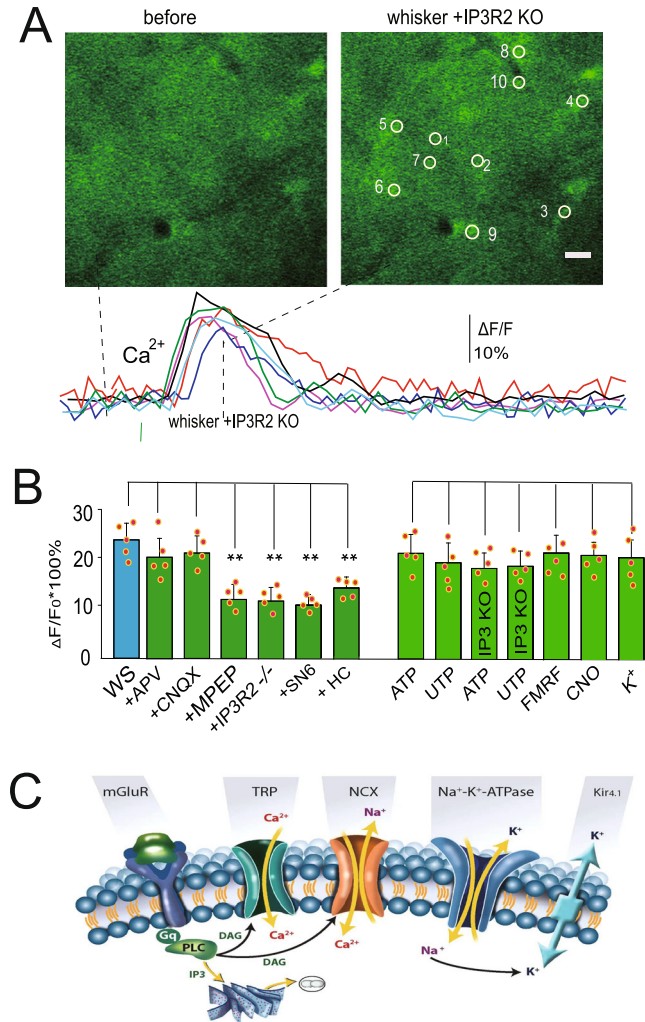

**Fig. 5 | Small Ca²⁺ transients during sleep are derived from multiple sources of Ca²⁺. A** Representative photomicrographs showing that whisker stimulation induced small Ca²⁺ transients in IP₃R₂ knockout mice, as illustrated by increased fluorescence intensity. The scale bar is 10 μm. The lower panel shows typical traces of Ca²⁺ transients. **B** Statistical data show the comparison of whisker stimulation-induced changes in Ca²⁺ transients with agonists and in IP3R₂ knockout mice with different manipulations of blockers (**$P < 0.001$, one-way ANOVA, $F_{(6, 232)} = 54.57$; spots show the averaged value in each mouse, $n = 5$ mice in each group, the data are shown as the mean ± SD). **C** A schematic diagram shows the intrinsic ion channels involved in the oscillation. Whisker stimulation-induced glutamate release from neurons affects astrocytes via mGluR (metabotropic glutamate receptor), and mGluR activates PLC (phospholipase C) to release IP3 (inositol trisphosphate) and DAG (diacylglycerol), which in turn activate TRP channels (transient receptor potential ion channels) and NCX (the Na⁺/Ca²⁺ exchanger) channels and thus Ca²⁺ influx. ATP, UTP, and FMRF in Mrg mice can also activate Gq receptors (G protein-coupled receptors) to activate PLC and release IP3 and DAG, thus inducing Ca²⁺ influx via TRP channels and NCX channels. Note: Kir4.1, an inwardly rectifying potassium (Kir) channel.

TTX-sensitive sodium channels in most astrocytes, except for those in some reactive astrocytes during injury[38], and few studies have reported sodium channels in astrocytes in the cortex[39–41]. Given these findings, we think that the amount of TTX applied in our experiments is unlikely to introduce uncontrolled issues that could potentially affect our conclusions.

To further test a potential mGluR mechanism, we examined Ca²⁺ transients with the calcium indicator Rhod-2-AM (10 μM) in mice with genetic deletion of the IP₃ receptor (Type 2). In the absence of the IP₃ receptor, whisker stimulation still induced Ca²⁺ transients in these mice

($17.5 ± 3.2$ %, $P < 0.05$, $n = 5$ mice, 36 ROIs, Fig. 5A), suggesting that during sleep, there is alternative Ca²⁺ entry that is independent of IP₃-dependent Ca²⁺ release from the ER. We hypothesized that these alternative Ca²⁺ transients may instead be due to the entrance of extracellular Ca²⁺ through transient receptor potential channels (TRPA1)[42,43] or through NCX[35]. To test this hypothesis, we administered the TRPA1 channel blocker HC030031 (100 μM) and found that whisker stimulation-induced Ca²⁺ transients were significantly decreased to $15.03 ± 3.1$% ($P < 0.05$, $n = 5$ mice, 36 ROIs, Fig. 5B). G protein (Gq/11) stimulation is known to induce phosphatidylinositol bisphosphate (PIP2) hydrolysis by phospholipase C (PLC) enzymes, which leads to activation of TRP channels via both IP3 and diacylglycerol (DAG)[44]. However, when we applied the NCX blocker SN-6 (50 μM) and SEA0400 (50 μM) on the surface of the brain, whisker stimulation-induced Ca²⁺ transients decreased (from $18.6 ± 3.7$% to $11.4 ± 3.09$%, $P < 0.01$, $n = 33$ cells in 5 mice, one-way ANOVA, Fig. 5B), suggesting that NCX also contributed to the sleep-associated small Ca²⁺ transients. One possibility was that this was due to activation by PLC enzyme-induced activation of IP3 and DAG, similar to previous reports that the TRP channel and NCX interaction played a role in DAG-dependent platelet aggregation[44]. Astrocytic processes contain many microdomains that contain NCX, which might work together with TRP channels to induce influx of Na⁺ and Ca²⁺. During sleep, extracellular Ca²⁺ increases[30], and intracellular Ca²⁺ is relatively low, so Ca²⁺ signaling is largely derived from both the extracellular space and ER (Fig. 5C).

### Small astrocytic Ca²⁺ transients during sleep may enable glymphatic K⁺ clearance

The data presented above provide compelling evidence for a direct role of astrocytic Ca²⁺ signaling in sleep and arousal brain states. Our screening of the whole-cell recordings of neurons revealed that after whisker stimulation, the membrane potentials hyperpolarized from $71.1 ± 2.1$ mV to $74.2 ± 2.5$ mV ($n = 5$ cells in 5 mice). Thus, we used agonists (CNO, FMRF, and UTP) to induce Ca²⁺ transients in the presence of TTX (1 μM) to block neuronal activity and found that these agonists induced similar hyperpolarization ($71.7 ± 2.1$ mV to $74.2 ± 2.5$ mV for CNO; $n = 5$ cells in 5 mice for each experiment, Fig. 6A, B). Extracellular K⁺ changes were measured after whisker stimulation-induced Ca²⁺ transients and revealed a decrease in extracellular K⁺ ($3.7 ± 0.4$ mM to $2.6 ± 0.2$ mM for UTP; $P < 0.05$, paired t-test, $n = 5$ cells in 5 mice for each experiment, Fig. 6C, D), suggesting that the hyperpolarization in neurons was due to decreased K⁺. In light of the suggestion that the glymphatic system is involved in K⁺ clearance[45], we assessed K⁺ clearance in aquaporin-4 (AQP4) knockout mice, whose glymphatic system had been suggested to be impaired, and found a reduction in the extent of decrease/clearance of K⁺. Thus, it appears that the small Ca²⁺ transients that are associated with sleep may serve to inhibit sensory information from causing arousal, and this appears to be mediated by NE released by the LC. Moreover, during this state, it appears that the astrocytic glymphatic system is enabled to perform the essential function of clearance. Such findings raise the provocative possibility that during sleep, astrocytes suppress sensory information from reaching the neocortex to enable the necessary removal of metabolic waste from the brain.

## Discussion

This study demonstrates that sensory inputs during sleep can induce deeper sleep, while sensory inputs during wakefulness can induce arousing wakefulness. Pharmacological manipulations supported the astrocytic Ca²⁺ in regulating these processes, which are regulated by noradrenergic input to the barrel cortex that serves this role in modulating different brain states. The interesting findings demonstrated that astrocytes in the barrel cortex mediate these effects by distinct large Ca²⁺ transients to enhance arousal, while small Ca²⁺ transients induce deeper sleep and squelch sensory input in this circuit (Fig. 7).

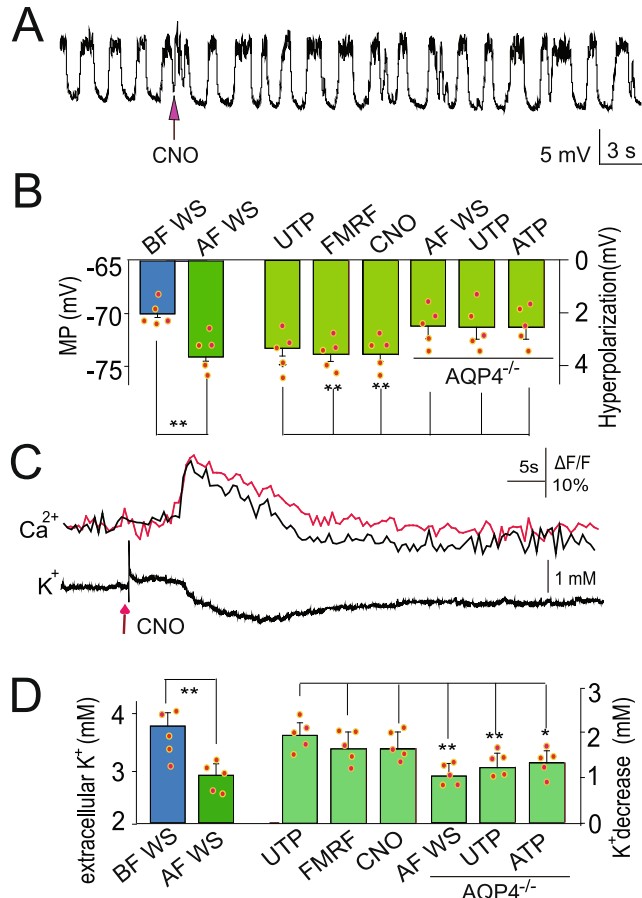

**Fig. 6 | Astrocytic Ca²⁺ transients enable glymphatic clearance of K⁺. A** Typical whole-cell recordings show that membrane potentials hyperpolarize after CNO-induced astrocytic Ca²⁺ transients. **B** Statistical analysis of whisker stimulation (WS) and agonist (UTP, FMRF, and CNO)-induced astrocytic Ca²⁺ transient-induced membrane hyperpolarization (**$P < 0.01$, the comparison between the left pair was done with paired t-test, $n = 5$; the comparison was done with one-way ANOVA among the other groups, $F(5,29) = 9.407$, $P < 0.001$; the spots show the value in each mouse, $n = 5$ in each group; the data are shown as the mean ± SD). **C** Typical traces show astrocytic Ca²⁺ transients and extracellular K⁺. **D** Statistical analysis of extracellular K⁺ changes due to whisker stimulation (WS) and agonist (UTP, FMRF, and CNO)-induced astrocytic Ca²⁺ transients (**$P < 0.01$, the comparison between the left pair was done with paired t-test, $n = 5$; the comparison was done with one-way ANOVA among the other groups, $F(5,29) = 13.87$, $P < 0.001$; the spots show the value in each mouse, $n = 5$ in each group, the data are shown as the mean ± SD).

Thus, we have found these distinct functional astrocytic roles within the complete circuitry and used live mice to elucidate this phenomenon, which has not been reported before.

The results from our study confirm a fundamental role for astrocytes as major players in the modulation of neurons under physiological conditions[10]. Beyond the local astrocytic control of synaptic activity by neurotransmitter uptake mechanisms, the role of astrocytic modulation in large-scale neuronal ensembles, such as global sleep and arousal states, has only recently been recognized[10]. For example, astrocytes were shown to facilitate sleep through adenosine[19] and connexin 43[19], by meeting metabolic demand[19], and by changing extracellular metabolites[46] or ion-associated mechanisms[3,47]. In addition, astrocytes were suggested to be involved in emotional arousal[48]. However, it needs to be emphasized that the two types of astrocytic Ca²⁺ signaling during the sleep state and during the arousal states have not been previously reported. Our study has made this valuable contribution to the understanding of astrocyte function in the somatosensory neocortex, and it would not be surprising to learn that other

areas of the neocortex also use astrocytic calcium waves as a sensory (and motor) gain.

Another important finding from these experiments is the fact that the LC and its neurotransmitter NE play a major role in modulating these behaviors by switching between small Ca²⁺ signaling during sleep to greater Ca²⁺ signaling in the arousal state. The latter LC/NE mechanism was previously suggested to play a role in modulating the brain arousal state[49,50]. The observations from the present study show that astrocytes are fundamentally and functionally involved in this modulation by enhancing sensory input and responsiveness during arousal and inhibiting sensory input during sleep. These findings expanded our understanding of the control of behavioral states.

In conclusion, by using integrative dual neuron and astrocyte recordings for local field potentials after NE or LC stimulation, we identified distinct roles of astrocytes in regulating sleep and arousal states, as demonstrated through the use of dual neuron and astrocyte recordings, as well as local field potentials and NE or LC stimulation[51]. Our results further reveal that astrocytic calcium waves play a key role in these processes. Sleep, wakefulness, and arousal are distinct patterns in the brain, each with its own unique neuromodulator profile and variations in brain wave synchronization, frequency, and amplitude[52]. These findings broadened our understanding of the unique contributions of astrocytes to the regulation of these behavioral states. Our study provided evidence that astrocytes not only promote sleep by silencing sensory transmission but also have the ability to enhance sensory transmission during wakefulness and arousal by increasing their activity, providing insights into the complex interplay between astrocytes and neurons and their impact on brain function.

## Methods

### Animal ethics statement

All animal procedures followed the National Institute of Health guidelines and were approved by the Institution of Animal Care and Use Committee at the University of Rochester (A3292-01) and the Institution of Animal Care and Use Committee at Sichuan Normal University (2021LS031).

### Animal preparation for awake in vivo recordings

Adult (10 weeks old) C57Bl/6 wild-type mice were used (both male and female, Charles River Laboratories). Mice were housed in a facility with a light/dark cycle of 12/12 h and provided food and water ad libitum. The preparation for mouse experiments was modified from published protocols. Briefly, mice were anesthetized using isoflurane (1.5% mixed with 1–2 L/min O₂), head restrained with a custom-made mini-frame and habituated to the restraint over one week in multiple sessions, with a total training duration of 3–4 h. A 1.5 mm craniotomy was then opened over the somatosensory cortex (1.5 mm in diameter, 3 mm lateral and 1.5 mm posterior to the bregma), the dura was carefully removed, and the mice were allowed 60 min of recovery prior to conducting the experiments. The craniotomy procedure lasted <20 min to minimize anesthesia exposure on the recording day. Mice were then head-strained, placed in a behavioral tube to minimize movement, and relocated to the imaging room, which was kept dark and asleep. The body temperature was maintained with a heating pad. For cortical drug surface application, artificial cerebrospinal fluid (aCSF) was perfused across the cortex of awake mice at a rate of 2 mL/min into a custom-made well with an ~200 µL volume through tubing with an ~100 µL volume, meaning the entire volume bathing the brain was exchanged every ~9 s. The aCSF solution contained 126 mM NaCl, 2.5 mM KCl, 1.25 mM NaH₂PO₄, 2 mM MgCl₂, 2 mM CaCl₂, 10 mM glucose, and 26 mM NaHCO₃, pH 7.4. For imaging, the calcium indicator rhod-2 AM (Invitrogen) was loaded onto the exposed cortex for 30–40 min before applying agarose (1.5%, type III-A, Sigma) and a

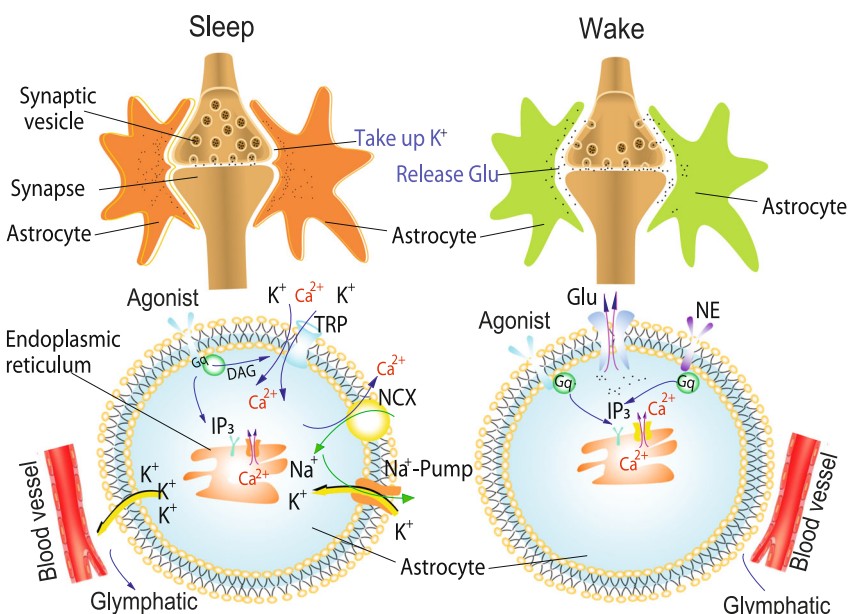

**Fig. 7 | A model shows the mechanisms of Ca²⁺ transients affecting EPSPs during sleep and waking states. Left:** During sleep, whisker stimulation induces glutamate release from neurons and activates mGluR (metabotropic glutamate receptor) and phospholipase C to release inositol trisphosphate (IP3) and diacylglycerol (DAG), which possibly open transient receptor potential ion channels (TRP channels) to obtain Ca²⁺ influx into astrocytes. The Ca²⁺ increase would in turn increase intracellular Na⁺, activating the Na⁺ pump to take up K⁺ and hyperpolarizing the neuron. **Right:** During waking states, norepinephrine (NE) activates G protein-coupled receptors (Gq receptors) that prime astrocytes and synergistically works with metabotropic glutamate (mGLu) to activate Gq receptors to release even more IP₃, which induces Ca²⁺ release from the endoplasmic reticulum (ER) and glio-transmitter release (such as ATP, glutamate, or glutamine) and induces emotional arousal states. Note: Glu glutamine; NCX, the Na⁺/Ca²⁺ exchanger or the sodium-calcium exchanger.

coverslip. All mice were trained several times a day for three days before the experiments. The mice were anesthetized with 1.5% isoflurane during the surgery, and then after being mounted on the equipment, the isoflurane was stopped and the mouse was left to run freely on the wheel during sleep/wake states in the dark room, except that the heads were held tightly on a frame.

## Chemicals
All standard chemicals were purchased from Sigma-Aldrich unless mentioned otherwise. CNQX (6-cyano-7-nitro-quinoxaline-2,3-dione, Sigma), clozapine-N-oxide (CNO) (Enzo Life Sciences), APV (D-2-amino-5-phosphono-pentanoic acid, Sigma), MPEP (6-(phenylethynyl)-pyridine, Tocris), terazosin (Tocris), metoprolol (Tocris), SN-6 (2-[4-(4-nitrobenzyloxy)benzyl]thiazolidine-4-carboxylic acid ethyl ester, Tocris), SEA0400 (2-[4-[(2,5-difluorophenyl)methoxy]-phenoxy]-5-ethoxyaniline, CM-4620 (Tocris), synthesized by Taisyo Pharmaceutical Co. Ltd), UTP (uridine 5′-triphosphate, Tocris).

## Surgical procedure for virus injection
pAAV5-GfaABC₁D-cyto-GCaMP6f-SV40 (Cat# AV-5-52925, UPenn vector core), which is a genetically encoded Ca²⁺ indicator driven by the astrocyte-specific GfaABC1D promoter, was injected into the barrel cortex. The mice were anesthetized with 1–2% isoflurane with oxygen supply and were placed in a stereotactic head frame with a heating pad underneath. A small vertical incision was made on the skin, and craniotomy (0.5 mm × 0.5 mm) was performed with a drill. The virus was injected at a volume of 500 μL/site without dilution, and 3 injection sites were utilized. A glass micropipette with a tip of 10 μm was used for injection with microinjection control.

## In vivo two-photon imaging and stimulation
A custom-built microscope attached to a Tsunami/Millenium laser (Spectra Physics, Mountain View, CA) and scan box (FV300 Fluoview Software Ver 4.3a, Olympus, Center Valley, PA) was used for 2-photon imaging through a 203 objective (0.9 NA, Olympus). The excitation

wavelength was in the range of 800–820 nm. Emission wavelengths were split to detect fluo-4 and AlexaFluor 594 signals as previously described[17]. Images of astrocytic Ca²⁺ signaling were recorded every 2–3 s, which was sufficient to capture evoked responses while limiting laser-induced photodamage at a laser power of <30 mW. Prior to whisker stimulation experiments, anesthesia 0.5 mg/kg D-tubocurarine was injected to prevent small reflex movements that could distort imaging. Direct LC stimulation was applied using a bipolar concentric electrode. Stimulation consisted of a single train of 20–100 pulses (100 Hz, 50 μA, 0.5 ms square pulses). ROIs were extracted from the above-threshold pixels with the fluorescence of the GCaMP or Rhod-2 imaging, and ΔF/F signals were calculated to detect the periods that had multiple ΔF/F peaks above baseline.

## In vivo whole cell recording
Recordings were obtained from the layer II barrel cortex using glass microelectrodes. LFP signals were externally filtered at 6 Hz (Filter Butterworth Model by Encore, Axopatch 200B by Axon Instruments), bandpass filtered at 1–100 Hz and digitized (Digidata 1440 A by Axon Instruments). Recordings were analyzed offline using pClamp 10.2. Whole-cell recordings were performed with blind patching by observing the pipette resistance. Patch electrodes were fabricated from filament thin-wall glass (World Precision Instruments) on a vertical puller; the resistance of the pipette was approximately 6 to 9 megohms with intracellular pipette solution added. The pipette solution contained 140 mM K-gluconate, 5 mM Na-phosphocreatine, 2 mM MgCl₂, 10 mM HEPES, 4 mM Mg-ATP, and 0.3 mM Na-GTP (pH adjusted to 7.2 with KOH). The junction potential between the patch pipette and the bath solution was zeroed before forming a gigaseal. Patches with seal resistances of less than 1 gigohm were rejected. Data were low pass–filtered at 2 kHz and digitized at 10 kHz with a Digidata 1440 interface controlled by pClamp Software (Molecular Devices). Whisker stimulation was delivered using a picospritzer III (Parken Instrumentation) and Master 8 (A.M.P.I.). The amplifier bandwidths were normally 0.5 Hz to 100 Hz. EEG recording was digitized at 100 Hz and

then subjected to spectral analysis using a complex demodulation procedure.

## EEG and EMG recordings

Acquired EEG/EMG signals were amplified (Filter Butterworth Model by Encore, Axopatch 200B by Axon Instruments) at a sampling frequency of 1 kHz. The EEG signal was filtered with high-pass: 0.5 Hz, low pass: 30 Hz, and EMG signal high-pass filtered at 10 Hz. Wakefulness was subdivided into quiet wakefulness (QW) or arousal wakefulness (AW) using the EMG peak-to-peak amplitude of all wake epochs across the 12-h recording. QW was defined as the 33rd percentile or less and AW as the 66th percentile or higher of all wake EMG peak-to-peak amplitude values. Concurrently with EEG recordings, spectral analysis of EEG recordings showed that most of its power resides in 4–6 Hz, which was interrupted intermittently with slow waves of 0.5–4 Hz.

## Statistical analysis

All analyses were performed using SPSS 19 software (IBM), and all tests were two-tailed where significance was achieved at the $\alpha = 0.01$ level. The data are shown as the mean ± S.D. (standard deviation). For independent samples, a $t$-test (≤2 variables) or one-way ANOVA (>2 variables) was used; for paired samples, a paired $t$-test was used.

## Reporting summary

Further information on research design is available in the Nature Portfolio Reporting Summary linked to this article.

## Data availability

All data needed to evaluate the conclusions in the paper are present in the paper and/or the Supplementary Information/Source data file. Data are also available upon request from the corresponding authors. Source data are provided with this paper.

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

## Acknowledgements

The paper was supported by the National Natural Science Foundation of China (82101602, F.W.; 82171392, S.G.), the Corbett Estate Fund (62285-531021-41800, E.W.), the Helen Vosburg McCrillus Plummer and Robert Edward Lee Plummer, Jr. Chair Fund (Jason Huang), and a grant from the National Institutes of Health (R01NS067435, J.H.H.). The authors would like to thank Dr. Hajime Hirase and Dr. Xuejing Wang for their valuable contributions and assistance with this study.

## Author contributions

F.W., E.W., and J.H.H. conceived, designed, and supervised the study; F.W., S.G., W.W., N.A.S., and W.P. performed the experiments; F.W., S.G., W.D., J.Y., D.Q., B.Z., Y.M., P.C., L.A.S., S.S.Y., E.W., and J.H.H. analyzed the data; F.W., S.G., E.W., and J.H.H. obtained funding; F.W. wrote the initial manuscript; D.Q., Q.R.L., L.A.S., S.S.Y., E.W., and J.H.H. substantially revised the manuscript. All authors read and approved the contents of the manuscript.

## Competing interests

The authors declare no competing interests.

## Additional information

¹Institute of Brain and Psychological Sciences, Sichuan Normal University, Chengdu, Sichuan 610060, China. ²Department of Neurosurgery, University of Rochester, Rochester, NY 14643, USA. ³Department of Biology, Boston University, Boston, MA 02215, USA. ⁴Department of Medical Psychology, Jiangsu University Medical School, Zhenjiang 212013, China. ⁵Department of Neurosurgery and Neuroscience Institute, Baylor Scott & White Health, Temple, TX 76508, USA. ⁶George Washington University School of Medicine and Health Sciences, Washington, DC 20052, USA. ⁷Center for Neuroscience Research, Children's National Research Institute, Children's National Hospital, Washington, DC 20010, USA. ⁸Key Laboratory of Medical Electrophysiology, Ministry of Education & Medical Electrophysiological Key Laboratory of Sichuan Province, Institute of Cardiovascular Research, Southwest Medical University, Luzhou 646000, China. ⁹Basic Medicine College, Hubei University of Chinese Medicine, Wuhan, China. ¹⁰Department of Neurosurgery, Huashan Hospital, Fudan University, Shanghai 200040, China. ¹¹School of Psychology, Nanjing University of Chinese Medicine, Nanjing 210023, China. ¹²Brain Tumor Center, Division of Experimental Hematology and Cancer Biology, Cincinnati Children's Hospital Medical Center, Cincinnati, OH 45229, USA. ¹³Department of Pediatrics, University of Cincinnati College of Medicine, Cincinnati, OH 45229, USA. ¹⁴Department of Neuroscience & Experimental Therapeutics, Texas A&M University,

Bryan, TX 77807, USA. [15]Livestrong Cancer Institutes and Department of Oncology, Dell Medical School, The University of Texas at Austin, Austin, TX 78712, USA. [16]Oden Institute for Computational Engineering and Sciences (ICES), The University of Texas at Austin, Austin, TX 78712, USA. [17]Department of Biomedical Engineering, Cockrell School of Engineering, The University of Texas at Austin, Austin, TX 78712, USA. [18]Texas A & M University Health Science Center, College Station, TX 77843, USA. [19]Department of Neurosurgery, Baylor College of Medicine, Temple, TX 76508, USA. [20]These authors contributed equally: Fushun Wang, Wei Wang, Simeng Gu, Dan Qi. ✉e-mail: 13814541138@163.com; lshapiro@tamu.edu; Stephen.yi@austin.utexas.edu; Erxi.Wu@bswhealth.org; Jason.Huang@bswhealth.org

