## [Peer Review File · Nature Communications]

Distinct astrocytic modulatory roles in sensory transmission during sleep, wakefulness, and arousal states in freely moving miceEditorial Note: This manuscript has been previously reviewed at another journal that is not operating a transparent peer review scheme. This document only contains reviewer comments and rebuttal letters for versions considered at Nature Communications .

Reviewers' Comments:

Reviewer #1:

Remarks to the Author:

The authors have addressed most of the comments, in particular by using TTX application to dissect the major involvement of neuronal activity. Although this reviewer is still uncertain about the conceptual advance involved or, otherwise, what burning questions this type of study answers, the methods and approaches employed are excellent and state-of-the-art. There are a couple of relatively minor issues that the authors should be able to address, as outlined below.

1. MPEP activates mGluR5 and therefore IP3R2, in addition to other alleged Ca²⁺ sources. Therefore, deletion of IP3R2 must reduce the overall mGluR5-driven Ca²⁺ signal, namely the part of it generated by IP3R2 - unless the Ca²⁺ -sensitive fluorescent signal is saturated across these trials. This discrepancy has to be explained.
2. TTX application may block TTX-sensitive sodium channels in astrocytes, with poorly controlled consequences. This has to be discussed.
3. Fig. 4 data: a similar issue to item 1 above. The well-established IP3R2-dependent Ca²⁺ mobilization component of mGluR2 activation cannot be simply ignored. What the authors suggest is that the mGluR2-dependent Ca²⁺ signals generated through IP3R2 are different from those generated through other sources. But both involve the same Ca²⁺ ions diffusing inside the same cell, and there is no evidence that they these two Ca²⁺ signals could be spatially separated. The complete lack of effect arising from IP3R2 deletion has to be explained. One notes, however, that the effects of IP3R2 deletion and BAPTA application seem to show a 'correct' trend in the summary bar graphs, albeit statistically insignificant for the present sample; this appears in line with some partial influence.

Reviewer #2:

Remarks to the Author:

This is an impressive set of experiments that could represent a paradigm shift in our understanding of astrocytic Ca²⁺ signaling. It is a carefully performed study that uses state-of-art techniques. The authors have addressed all my critiques and I have no additional questions.

A point-by-point response to all reviewers' comments

Note: comments are in *italics* and answers are in blue.

Reviewer #1 (Remarks to the Author):

The authors have addressed most of the comments, in particular by using TTX application to dissect the major involvement of neuronal activity. Although this reviewer is still uncertain about the conceptual advance involved or, otherwise, what burning questions this type of study answers, the methods and approaches employed are excellent and state-of-the-art. There are a couple of relatively minor issues that the authors should be able to address, as outlined below.

We greatly appreciate your positive and thoughtful evaluation. By addressing the critiques from you and other reviewers on the early version of our manuscript, the revised manuscript has been significantly improved.

By reviewing the other reviewer's comments below and our summary in the Discussion of the manuscript, we would like to further clarify our study's conceptual advance. Our studies address the distinct roles of astrocytes in regulating sensory transmission during varying states of sleep, wakefulness, and arousal, which have not been reported previously. Our studies uncover the unique contributions of astrocytes in shaping neural activity and sensory processing during these states. We have revised the Discussion section to clarify the novelty and significance of our study.

We have included the new changes here **“In conclusion, by using integrative dual neuron and astrocyte recordings for local field potentials after NE or LC stimulation, we identified distinct roles of astrocytes in regulating sleep and arousal states, as demonstrated through the use of dual neuron and astrocyte recordings, as well as local field potentials and NE or LC stimulation. Our results further reveal that astrocytic calcium waves play a key role in these processes. Sleep, wakefulness, and arousal are distinct patterns in the brain, each with its own unique neuromodulator profile and variations in brain wave synchronization, frequency, and amplitude (Liang et al, 2021). These findings represent a paradigm shift in our understanding of the unique contributions of astrocytes to the regulation of these behavioral states. Our study provided evidence that astrocytes not only promote sleep by silencing sensory transmission but also have the ability to enhance sensory transmission during wakefulness and arousal by increasing their activity, providing insights into the complex interplay between astrocytes and neurons and their impact on brain function.”**

Additionally, we have improved the clarity and concision of the Abstract.

We appreciate the opportunity to address the remaining minor concerns raised by you as we are committed to ensuring the highest possible quality of our paper.

References for this question:

Liang F, Xu Q, Jiang M, Feng R, Jiang S, Yuan B, Xu S, Wu T, Wang F, Huang JH. Emotion Induced Monoamine Neuromodulator Release Affects Functional Neurological Disorders. *Frontiers in Cell and Developmental Biology*. 2021. 9, 633048.

1. MPEP activates mGluR5 and therefore IP3R2, in addition to other alleged Ca²⁺ sources. Therefore, deletion of IP3R2 must reduce the overall mGluR5-driven Ca²⁺ signal, namely the part of it generated by IP3R2 - unless the Ca²⁺ -sensitive fluorescent signal is saturated across these trials. This discrepancy has to be explained.

Thank you for your careful review of our manuscript and for providing helpful suggestions for improvement. We appreciate your comment regarding the limited effectiveness of MPEP in inhibiting astrocytic mGluRs, given that it selectively targets mGluR5. This results in a smaller inhibition compared to IP3R2 KO, as shown in Figure 2D. To address this issue, we used a combination of MPEP (50 μ M, 2-methyl-6-(phenylethynyl)-pyridine), which targets mGluR5, and LY341495 (10 μ M), an mGlu_{2,3} receptor antagonist. This combination resulted in inhibition levels similar to those achieved through IP3R2 KO, as shown in Figure 3B.

Per your suggestions, we have now added a discussion on this discrepancy to the revised manuscript as follows:

“The characteristics of astroglial Ca²⁺ signals induced by whisker stimulation were examined in the presence of two antagonists: mGlu5 receptor antagonist MPEP (50 μ M) and mGlu_{2,3} receptor antagonist LY341495 (10 μ M). The resulting Ca²⁺ signal, $\Delta F/F_0$, decreased to $19.2 \pm 6.6\%$ during waking states (56 ROIs in 5 animals), as shown in Figure 3B. However, when MPEP (50 μ M) alone was used, the decrease in $\Delta F/F_0$ was not significant (Figure 2D). This finding is consistent with a previous report indicating that the expression of mGluR5 in astrocytes is developmentally regulated and undetectable after postnatal week 3, whereas mGluR3 is expressed in astrocytes at all developmental stages (Sun et al., 2013). In addition, other studies have also reported on the effects of mGlu_{2,3} receptors in astrocytes (Copeland and Salt, 2022).”

References for this question:

Sun W, McConnell E, Pare JF, Xu Q, Chen M, Peng W, Lovatt D, Han X, Smith Y, Nedergaard M. Glutamate-dependent neuroglial calcium signaling differs between young and adult brain. *Science*. 2013 Jan 11;339(6116):197-200.

Copeland CS, Salt TE. The role of thalamic group II mGlu receptors in health and disease. *Neuronal Signaling*. 2022 Nov 15;6(4):NS20210058.

2. TTX application may block TTX-sensitive sodium channels in astrocytes, with poorly controlled consequences. This has to be discussed.

We have found, both from our own experience and from the literature (Thio & Sentheimer, 1993; Verkhratsky & Nedergaard, 2016; Pappalardo et al., 2017; Zhang et al., 1996), that there are very few TTX-sensitive (TTX-S) sodium channels in astrocytes, with the exception of some reactive astrocytes during injury. Furthermore, only a handful of studies have reported Na channels in astrocytes in the cortex. Given these findings, we believe that the amount of TTX applied in our experiments is unlikely to introduce uncontrolled issues that could potentially affect our conclusions.

We have added a discussion on this in the revised manuscript to improve the clarity and readability of this manuscript.

References for this question:

Thio CL, Sontheimer H. Differential modulation of TTX-sensitive and TTX-resistant Na⁺ channels in spinal cord astrocytes following activation of protein kinase C. *The Journal of neuroscience*. 1993. 13(11), 4889–4897.

Pappalardo LW, Black JA, Xaxman SG. Sodium channels in astroglia and microglia. *Glia*. 2017. Oct; 64(10): 1628–1645.

Verkhratsky A, Nedergaard M. Physiology of Astroglia. *Physiological Reviews*. 2018 Jan 1;98(1):239-389.

Zhang X, Phelan KD, Gelller HM. A novel tetrodotoxin-resistant sodium current from an immortalized neuroepithelial cell line. *Journal of Physiology*. 1996, 490.1, pp.17-29.

3. Fig. 4 data: a similar issue to item 1 above. The well-established IP3R2-dependent Ca²⁺ mobilization component of mGluR2 activation cannot be simply ignored. What the authors suggest is that the mGluR2-dependent Ca²⁺ signals generated through IP3R2 are different from those generated through other sources. But both involve the same Ca²⁺ ions diffusing inside the same cell, and there is no evidence that they these two Ca²⁺ signals could be spatially separated. The complete lack of effect arising from IP3R2 deletion has to be explained. One notes, however, that the effects of IP3R2 deletion and BAPTA application seem to show a 'correct' trend in the summary bar graphs, albeit statistically insignificant for the present sample; this appears in line with some partial influence.

Thank you for bringing up the importance of mGluR2 in our study. We completely agree with you that mGluR2-dependent Ca²⁺ signals play a critical role in astrocytes. In this study, we employed an mGluR2 antagonist, LY341495 (10 μM), in our experiments. As shown in Figure 3B, this addition has significantly enhanced the inhibition of Ca²⁺ signaling.

Reviewer #2 (Remarks to the Author):

This is an impressive set of experiments that could represent a paradigm shift in our understanding of astrocytic Ca²⁺ signaling. It is a carefully performed study that uses state-of-art techniques. The authors have addressed all my critiques and I have no additional questions.

We greatly appreciate your time and thoughtful evaluation. By addressing the critiques from you and other reviewers on the early version of our manuscript, the revised manuscript has been significantly improved.

Reviewer #1 additional comments:

1. Any negative results in pharmacological dissection tests in vivo require positive controls. When a ligand application has no effect on a selected physiological readout, one has to test if

the application protocol actually reaches its targets: rapid washout of drugs in vivo is a common scenario. This relates to the critical results following the application of TTX, BAPTA, BaCl₂ (no effect on neurons?), glutamate receptor blockers, etc.

Thank you for your comment. We agree that TTX, BAPTA, and BaCl₂ can all have effects on neurons. However, in our study, we used TTX (1 μM) exclusively in the context of agonist-induced Ca²⁺ waves, where we could exclude any involvement of neurons. The application of TTX, along with the agonist resulted in the observation of agonist induced Ca²⁺ signaling, suggesting that TTX reached the tissue and exerted their effects. The experiments with BaCl₂ and BAPTA which were included in the initial submission to Nature have been removed after revision.

2. Selection of ROIs in Ca²⁺ imaging, which is a critical step, appears fairly arbitrary. The criterion of having higher baseline fluorescence and larger fluorescence responses suggests biased, rather than randomised sampling. The cytosolic GCaMP6f signal at rest will scale with the local astrocyte volume, so the proposed criterion is skewed towards larger, rather than smaller, cell branches. Overall, it would seem more appropriate to image the entire astrocyte areas representing nanoscopic protrusions, in a good number of astrocytes, to get an unbiased readout of Ca²⁺ signal under different conditions.

While it would be ideal to image the entire astrocyte, including their nanoscopic protrusions, it is challenging to differentiate astrocytes from neutrophils. Furthermore, astrocytic Ca²⁺ changes typically occur in specific regions of the cell known as "buttons" rather than uniformly throughout the cell. As a result, many studies focus only on examining Ca²⁺ spots, especially for many recent studies, which only studied the fine processes which are loaded with GCaMP6f vectors (for example: Monai et al. 2016). In Figure 5A, we present an example of the region of interest (ROI) with the strongest Ca²⁺ signal. To obtain a representative measurement of the average Ca²⁺ activity in the entire astrocyte, we selected multiple ROIs from the image.

Reference for this question:

Monai H, Ohkura M, Tanaka M, Qe Y, Konno A, Hirai H, Mikoshiba K, Itohara S, Nakai J, Iwai Y, Hirase H. Calcium imaging reveals glial involvement in transcranial direct current stimulation-induced plasticity in mouse brain. **Nature Communications**. 2016 Mar 22;7:11100.

3. The authors tend to refer to published observations as a methodological justification for their estimates. Unfortunately, given no unified concept about the principles of astroglial signalling, the astrocyte literature often suffers from misinterpretation of Ca²⁺ imaging readout. In most cases, the reported claims are irreproducible, if only due to the multifactorial complexity of experiments in individual labs. It is therefore important to have all control tests at hand rather than through a reference.

Thank you for your suggestion. We acknowledge that some published observations may not be reliable, even in high-impact journals, and therefore, it is crucial to not rely solely on reported methods. To ensure the robustness of our own experiments, we exercised caution and incorporated appropriate controls when testing drugs. Additionally, we referred to published observations, which are in keeping with our experimental results, while keeping in mind the requirement for independent verification.

Reviewers' Comments:

Reviewer #1:

Remarks to the Author:

The authors have made an effort in addressing the remaining comments. This reviewer still does not understand what 'paradigm' has been shifted; in other words, which long-established general concept, if any, has been rejected in favor of the new one. I would suggest the authors to tone down such claims.

A point-by-point response to all reviewers' comments

Note: comments are in *italics* and answers are in blue.

Reviewer #1 (Remarks to the Author):

The authors have made an effort in addressing the remaining comments. This reviewer still does not understand what 'paradigm' has been shifted; in other words, which long-established general concept, if any, has been rejected in favor of the new one. I would suggest the authors to tone down such claims.

We would like to express our sincere gratitude to the reviewer for taking the time to provide us with valuable feedback. We greatly appreciate the insightful comments, which we have carefully considered in revising our manuscript. We have addressed the concern raised by the reviewer regarding our use of the phrase “paradigm shift” and have made the necessary changes to our manuscript. Although it is worth noting that one of the reviewers had used this phrase to praise our findings, we have removed the final sentence of the abstract to avoid making any exaggerated claims. Once again, we thank the reviewer for the valuable input.